# Reinforcement Learning with Task Decomposition and Task-Specific Reward System for Automation of High-Level Tasks

**DOI:** 10.3390/biomimetics9040196

**Published:** 2024-03-26

**Authors:** Gunam Kwon, Byeongjun Kim, Nam Kyu Kwon

**Affiliations:** Department of Electronic Engineering, Yeungnam University, Gyeongsan 38541, Republic of Korea; nineman@yu.ac.kr (G.K.); slim7928@ynu.ac.kr (B.K.)

**Keywords:** deep reinforcement learning, soft actor–critic, task decomposition, high-level task

## Abstract

This paper introduces a reinforcement learning method that leverages task decomposition and a task-specific reward system to address complex high-level tasks, such as door opening, block stacking, and nut assembly. These tasks are decomposed into various subtasks, with the grasping and putting tasks executed through single joint and gripper actions, while other tasks are trained using the SAC algorithm alongside the task-specific reward system. The task-specific reward system aims to increase the learning speed, enhance the success rate, and enable more efficient task execution. The experimental results demonstrate the efficacy of the proposed method, achieving success rates of 99.9% for door opening, 95.25% for block stacking, 80.8% for square-nut assembly, and 90.9% for round-nut assembly. Overall, this method presents a promising solution to address the challenges associated with complex tasks, offering improvements over the traditional end-to-end approach.

## 1. Introduction

Robot manipulators are characterized by high efficiency and accuracy in performing repetitive and precise tasks, thereby augmenting their considerable potential for application across various fields such as manufacturing, logistics, and service industries [1]. Robot manipulators, constructed with a structure resembling that of the human arm, consist of joints like wrists and elbows, which are interconnected by links, enabling them to execute movements similar to those of humans [2]. The structural design of these manipulators enables a range of motion and rotation capabilities, providing both flexibility and precision. These attributes allow robot manipulators to have the ability to adapt to a wide range of human behaviors and demands. This capability is considered an essential characteristic in the field of human–robot interaction (HRI) [3]. The properties of robot manipulators can complement human work abilities and contribute to increasing not only efficiency but also safety in more complex and diverse work environments.

In general, robots exhibit superior abilities to humans in repetitive and precise tasks, whereas humans are capable of comprehensive thinking and judgment [4]. Consequently, collaboration between humans and robots not only compensates for the weaknesses of each but also maximizes their strengths, leading to mutual synergistic effects that extend the application range of robots and enhance work efficiency [5]. To enhance this human–robot synergy effect, continuous research and development has been studied, resulting in the emergence of the concept of collaborative robots (cobots) and their expansion into various fields. Unlike traditional industrial robots that are primarily used in automated manufacturing and logistics industries, cobots are increasingly being applied to a wider range of tasks due to their capability to execute high-difficulty operations [6]. Therefore, cobots are expanding into diverse fields such as food and beverages, service, and healthcare, becoming commonplace in daily life in various forms such as robots that make coffee, open doors, or assist in medical surgeries [7,8]. Through these cases, interaction between humans and robots in unstructured environments is recognized as an important issue [9]. To implement effective interaction, it is necessary to develop various robot control algorithms that are customized to specific tasks and situations.

Meanwhile, control techniques based on system models, such as linear quadratic control [10], sliding mode control [11], computed-torque control [12], and model predictive control [13], allow for the precise prediction and control of robot movements. This requires the mathematical modeling of the system’s physical and mechanical characteristics and tuning of parameters like control gains. However, it is very difficult to build a perfect mathematical model that considers all situations and the physical and dynamic characteristics of the system, and parameter tuning for optimal control performance not only takes a lot of time but also has the possibility of failure. To overcome these limitations, various control techniques, including direct teaching, teleoperation, and reinforcement learning, have been investigated. Direct teaching is a method where the user manually manipulates the position or path of the robot arm to teach the robot the desired task [14]. Teleoperation technology utilizes various interfaces such as keyboards, joysticks, and touchscreens to control the robot [15]. The advantage of these approaches is that it does not require the mathematical modeling of the system or gain tuning. However, while control methods through direct teaching or teleoperation can be effective for simple tasks, they may have limitations for complex tasks or unexpected situations, potentially requiring additional programming or human intervention [16]. To address these limitations, researchers are trying to give robots the ability to think and act on their own with active decision making. For example, technology has been developed to enable robots to make flexible and intelligent decisions, such as responding to environmental changes, recognizing obstacles, and automatically adjusting their paths as needed. In particular, reinforcement learning algorithms are one of the primary methods to achieve this objective and have recently received significant attention among researchers.

Reinforcement learning is a machine learning technique that trains an agent through trial and error to actively decide on actions that maximize cumulative rewards within an environment [17]. Through this technique, the agent learns to efficiently respond and adapt to achieve its goals in various situations. This approach is similar to how humans learn, where the agent updates its behavior through trial and error and ultimately establishes an action policy that maximizes cumulative rewards [18]. This process enables the agent to robustly respond to various environmental changes and acquire the ability to actively recognize and solve problems like humans [19]. Moreover, since reinforcement learning is achieved through interaction with the environment, it has the advantage of not requiring gain tuning or mathematical modeling. Based on these characteristics of reinforcement learning, del Real Torres et al. [20] suggest that there is potential and promise for the application of reinforcement learning algorithms in automation fields, particularly in smart factories and robotics. Despite these advantages, reinforcement learning still faces challenges in terms of adapting to complex environments, ensuring stability, and maintaining predictability, especially in complex tasks such as block stacking and pick and place during robot manipulator tasks [21].

The Hierarchical Reinforcement Learning (HRL) approach is one of the methods to overcome these challenges. HRL utilizes a hierarchical structure in the learning process to effectively solve complex and high-dimensional tasks [22]. The authors of [23] utilized the HRL approach, which divides pick-and-place tasks into three subtasks: approach, manipulate, and retract. These three subtasks are trained by the Deep Deterministic Policy Gradient (DDPG) algorithm [24] with the Hindsight Experience Replay (HER) technique [25] and are coordinated using a High-Level Choreographer (HLC). Through this, pick-and-place tasks are successfully performed. Additionally, another research article [26] decomposes pick-and-place tasks into two reaching tasks and one grasping task. The two reaching tasks are trained using soft actor–critic (SAC), and the grasping task is implemented with a simple method of applying force to the gripper. Afterward, the pick-and-place task is successfully performed by sequentially connecting the subtasks: reaching, grasping, and reaching. Previous research applied task decomposition techniques and a task-specific reward system only to pick-and-place tasks among high-level tasks. To the best of our knowledge, no existing research has demonstrated a high success rate in the performance of the other high-level tasks through the application of task decomposition technology and a task-specific reward system, which motivates this article.

This paper proposes a reinforcement learning method based on task decomposition and a task-specific reward system for performing tasks more complex than the pick-and-place task, such as door opening, block stacking, and nut assembly. Initially, the door-opening task is subdivided into the processes of reaching–grasping–turning–pulling, the block-stacking task into reaching–grasping–reaching–putting, and the nut assembly task into reaching–aligning–reaching–grasping–assembling–putting. Among the subdivided subtasks, the grasping and putting tasks are implemented through single joint and gripper actions, while the remaining tasks establish optimal policies through the SAC algorithm and a task-specific reward system. Here, the task-specific reward system is used to increase the training speed of the agent, improve success rates, and facilitate the subsequent tasks of grasping and putting. Subsequently, by sequentially connecting the established policies, we confirm the successful execution of high-level tasks such as door opening, block stacking, and nut assembly. This demonstrates that the proposed method overcomes the limitations of the end-to-end approach and represents a useful solution for solving various complex and challenging tasks.

## 2. Related Work

### 2.1. Hierarchical Reinforcement Learning

HRL is an approach that utilizes a hierarchical structure in the training process to effectively address complex and high-dimensional tasks within the field of reinforcement learning. Traditional reinforcement learning involves an agent interacting with the environment and updating its action policy in response to rewards or penalties received as feedback. However, in complex tasks, the high dimensionality of the problem makes it challenging for agents to train and find the optimal policy. HRL addresses these issues by introducing a hierarchical structure into the decision-making process of the agent. The main idea of HRL is that, instead of learning a single complex policy for the entire task, it focuses on learning sets of subpolicies for specific aspects or subtasks. These subpolicies are organized in the hierarchical method, enabling more efficient learning and decision making. In summary, HRL is a more effective approach than traditional reinforcement learning by utilizing structured decision-making layers to solve problems arising in complex, high-dimensional tasks.

### 2.2. Soft Actor–Critic

SAC is an off-policy, model-free approach in reinforcement learning. This algorithm focuses on maximizing cumulative rewards while learning stochastic policies in continuous state space. It particularly shows outstanding performance in training agents with high-dimensional states and continuous action space, such as robotic arms and autonomous driving. Furthermore, by incorporating the maximum entropy method and soft Q-functions, it enables exploration to occur effectively for a variety of experiences in uncertain environments. The core principle of maximum entropy RL is to integrate the concept of entropy into the decision-making process. Therefore, maximum entropy RL aims to establish policies that not only maximize cumulative rewards over time but also ensure a diverse distribution of actions in each state. This approach achieves a harmonious balance between exploration and exploitation based on the state. A policy optimization formula of SAC is represented as follows:(1)$$\pi^{*}=\underset{\pi }{\mathrm{argmax}}\sum_{t}E_{{(s}_{t}{,a}_{t}{)\sim \rho }_{\pi }}\left[r\left(s_{t}{,\;a}_{t}\right)+\alpha H\left(\pi \left(\;\cdot {|s}_{t}\right)\right)\right],$$
where $r\left(s_{t}{,\;a}_{t}\right)$ denotes an immediate reward function when the state is $s_{t}$ and action is $a_{t}$. For a given policy π  and state $s_{t}$, the term $H\left(\pi \left(\;\cdot {|s}_{t}\right)\right)$ denotes the entropy that encourages the exploration. α is the temperature parameter that adjusts the level of randomness in the chosen policy and reflects the relative importance of the entropy component in the overall structure of the policy. This approach is adjusted according to the reward context of each state: in states with high rewards, policies with low entropy (i.e., more predictable actions) are considered sufficient, while in states with low rewards, policies with high entropy (i.e., more exploratory actions) are preferred to encourage broader exploration. Moreover, SAC employs a soft Q-function to facilitate exploration and prevent the policy from becoming overly biased, thereby adding flexibility to the decision-making process. The SAC algorithm utilizes an actor–critic structure, where the actor decides the actions of the agent based on the current state, and the critic evaluates the action determined by the current state. In the SAC research in [27], the actor and critic are implemented as deep neural networks, with the policy network functioning as the actor and the soft Q-network serving as the critic. The objective function of the critic network is defined using mean squared error (MSE) between the Q-value and the target Q-value, with the goal of minimizing this MSE. The Q-value is estimated through the soft Q-network and the target Q-value is approximated through the target soft Q-network. The formula is as follows:(2)$$J_{Q}\left(\theta \right){=E}_{\left(s_{t}{,\;a}_{t}\right)\sim D}\left[\frac{1}{2}{\left(Q_{\theta }\left(s_{t}{,\;a}_{t}\right)\;-\;(Q_{\bar{\theta }}\left(s_{t}{,a}_{t}\right))\right)}^{2}\right],$$

Furthermore, the parameters of the soft Q-network are denoted by θ, and the parameters of the target soft Q-network are represented by $\bar{\theta }$. The parameters $\bar{\theta }$ is updated by copying from θ at fixed intervals, a process that enhances the stability of the critic network’s training. Additionally, optimization of θ employs the stochastic gradient descent (SGD) method to minimize the Bellman residual. The gradient of the objective function is expressed as follows:(3)$${\hat{\nabla }}_{\theta }{\;J}_{Q}\left(\theta \right)=\nabla_{\theta }Q_{\theta }{(s}_{t}{,a}_{t}{)(Q}_{\theta }\left(s_{t}{,a}_{t}\right)\;-\;(r\left(s_{t}{,a}_{t}\right){+\gamma (Q}_{\bar{\theta }}\left(s_{t+1}{,a}_{t}\right)\;{-\;\alpha \log(\pi }_{\phi }{(a}_{t+1}{|s}_{t+1})))).$$

Next, the objective function of the actor network is expressed as Equation (4), based on the Kullback–Leibler divergence (KLD) between the entropy of the policy and the Q-value, where ϕ represents the parameters of the actor network.
(4)$$J_{\pi }\left(\phi \right){=E}_{s_{t}\sim D}{[E}_{a_{t}{\sim \pi }_{\phi }}{[\alpha \log(\pi }_{\phi }{(a}_{t}{|s}_{t}{)\;-\;Q}_{\theta }{(s}_{t}{,a}_{t})]].$$

Before the gradient is calculated, the objective function of the actor network is modified through the reparameterization trick, represented as $a_{t}{=f}_{\phi }(\epsilon_{t}{;s}_{t})$. The policy of SAC leverages a differentiable Q-function as its target, thus enabling the attainment of low variance through the reparameterization trick. This method significantly enhances the convergence speed of policy. Furthermore, utilizing this trick leads to the reparameterization of the actor’s objective function as the following form:(5)$$J_{\pi }\left(\phi \right){=E}_{s_{t}\sim D,\epsilon_{t}\sim N}{[\alpha \log\pi }_{\phi }{(f}_{\phi }(\epsilon_{t}{;s}_{t}{)|s}_{t}{)\;-\;Q}_{\theta }{(s}_{t}{,f}_{\phi }\left(\epsilon_{t}{;s}_{t}\right))],$$

Then, the actor network is updated through SGD to minimize the KLD between the entropy of the policy and the Q-values. The gradient of the objective function is as follows:(6)$${\hat{\nabla }}_{\phi }{\;J}_{\pi }\left(\phi \right)=\nabla_{\phi }{\alpha \log(\pi }_{\phi }{(a}_{t}{|s}_{t}))+(\nabla_{a_{t}}\alpha \log\left(\pi_{\phi }\left(a_{t}{|s}_{t}\right)\right)\;-\;\nabla_{a_{t}}{Q(s}_{t}{,a}_{t}))\nabla_{\phi }f_{\phi }(\epsilon_{t}{;s}_{t}).$$

Through this process, the agent establishes an optimal policy that maximizes both rewards and entropy by utilizing experiences gained from the variety of actions.

## 3. Proposed Method: Task Decomposition for High-Level Task

Task decomposition is a technique to simplify complex high-level tasks by dividing them into simpler and easier low-level tasks. In the Robosuite environment, high-level tasks such as door opening, block stacking, and nut assembly are decomposed into subtasks, and SAC is utilized to train agents to establish optimal policies for each decomposed subtask. Afterwards, the established subpolicies are modularized and connected sequentially to enable efficient performance of complex tasks. This method allows for the experimental validation of the performance of the agent. All tasks begin training from an initial state as shown in Figure 1.

### 3.1. Door Opening

The door-opening task is one of the common benchmark tasks in the fields of robotics and automation. The environment for the door-opening task is designed around a scenario in which a robot manipulator interacts with a door handle to open the door. The goal of this task is to enable the robot manipulator to accurately grasp the door handle and successfully open the door. In this paper, the door-opening task is divided into the following four subtasks.

Reaching task: reaching the door handle;Grasping task: grasping the door handle;Turning task: turning the door handle to unlock the door;Pulling task: pulling the door open while holding the handle.

The door-opening task is performed by sequentially linking the four tasks. Except for the grasping task, the reaching, turning, and pulling tasks are trained using the SAC algorithm. The first reaching agent is trained to perform the end effector reaches the position of the door handle, which is generated randomly. The agent of the grasping is designed to perform the end effector grasps of the handle through a single gripper action without any training. The turning agent is trained to release the lock by turning the handle. Finally, the agent for the pulling is trained to pull the door. Figure 2 shows the implementation process of the door-opening task using the proposed method.

### 3.2. Block Stacking

Block stacking is the task of stacking or loading blocks. It is a major task that can be used in various application fields such as manufacturing, logistics, and construction. The environment for the block-stacking task includes a scenario where the robot manipulator interacts with multiple blocks to stack them. The objective is for the robot manipulator to accurately manipulate and stack the blocks at the target position. In this research, the block-stacking task is divided into the following four subtasks.

Reaching task: reaching the target block;Grasping task: grasping the target block;Reaching task: stacking the grasped target block on top of the base block;Putting task: putting the target block and checking the state of the stacked blocks.

The block-stacking task is implemented by sequentially linking four subtasks. During this process, two reaching agents are trained using the SAC algorithm. The agent designed for grasping is configured to enable the end effector to grasp the target block using a single gripper action, without any training. The second reaching agent is trained to reach the grasped target block to the target position, which is on top of the base block. The putting agent is designed to check the stacked state of target blocks with single joint and gripper actions, not requiring any training. Figure 3 illustrates the implementation process of the block-stacking task using the proposed method.

### 3.3. Nut Assembly

The nut assembly task is the main task for the mechanical automation of robot manipulators. The environment of the nut assembly task includes a robot manipulator, nuts, and pegs that the nuts are to be fitted onto. The goal of this task is for the robot manipulator to accurately grasp the nut, move it, and align it with the peg. Through the proposed methodology, nut assembly can be divided into six subtasks as follows:Reaching task: reaching above the nut handle;Aligning task: aligning the orientation of the nut and the end effector;Reaching task: reaching the handle of the nut;Grasping task: grasping the handle of the nut;Assembly task: assembling the nut onto the peg;Putting task: checking the peg-in-hole state.

The nut assembly task is implemented by sequentially linking these subtasks. Additionally, all subtasks except for the grasping and putting tasks are trained using SAC. The first reaching agent is trained to reach the target position set directly above the handle of the nut. The aligning agent is trained to adjust the orientation of the end effector and the nut. The second reaching agent is trained to reach the handle of the nut. The agent for the grasping is configured to enable the end effector to grasp the handle using a single gripper action, without the need for any training. The assembling agent is trained to precisely peg-in-hole the nut onto the peg. Finally, the putting agent is designed to check the state of the assembled nut through single joint and gripper actions, without any training. Figure 4 shows the process of implementing the nut assembly task using the proposed method.

### 3.4. Designing States, Actions, and Task-Specific Reward System

#### 3.4.1. States and Actions

In this subsection, we describe the state and action for each task. Table 1 details the state of the robot manipulator and the state of the object for each task. In this paper, the Panda robot manipulator is used as the agent in all experimental environments. Since the Panda robot manipulator consists of seven joints and one gripper, the action $a_{t}\;\in {\;R}^{8}$ is defined for all tasks as in Equation (7) and represented by eight torques $\tau_{i}$ as in Equation (8). The direction of motion for each joint is determined by the sign of its action value. A positive action value indicates counterclockwise rotation and a negative value indicates clockwise rotation. Similarly, the action value of the gripper is categorized into opening and closing operations based on its sign. A positive action value corresponds to a closing operation, while a negative value corresponds to an opening operation. Additionally, the number of states varies depending on the task, with 46 for door opening, 51 for block stacking, and 57 for nut assembly.
(7)$$a_{t}{=[\tau }_{1_{t}}{,\tau }_{2_{t}}{,\tau }_{3_{t}}{,\tau }_{4_{t}}{,\tau }_{5_{t}}{,\tau }_{6_{t}}{,\tau }_{7_{t}}{,\tau }_{8_{t}}],$$
(8)$$-1\;\leq {\;\tau }_{i}\;\leq \;1,\;\forall i\in \;1,\;2,\;\cdots ,\;8.$$

First, the definition of the state that is commonly utilized across all tasks is as follows: the state of the robot manipulator’s end effector, which is denoted as $S_{\mathrm{ee}}=\left[P_{\mathrm{ee}}{,Q}_{\mathrm{ee}}\right]\;\in {\;R}^{7}$, comprises its current position and orientation. Here, the position is represented by $P_{\mathrm{ee}}\;\in {\;R}^{3}$, and the orientation is expressed in quaternion form as $Q_{\mathrm{ee}}\;\in {\;R}^{4}$. The cosine and sine values of the robot manipulator’s joint angles are represented by $\Theta_{\cos}\;\in {\;R}^{7}\;$ and $\Theta_{\sin}\;\in {\;R}^{7}$. The joint velocities are denoted by $\Theta_{\mathrm{vel}}\;\in {\;R}^{7}$. Lastly, the position and velocity of the manipulator’s gripper are indicated by $S_{g}\;\in {\;R}^{4}$. In this study, both the initial position of the robot manipulator and the position of the generated object are defined within the Cartesian coordinate system.

Next, the definition of the state used for each task is as follows: First, in the door-opening task, Ph ∈ R3 denotes the position of the door handle that the robot manipulator aims to reach. The positional difference between the end effector and the door handle is denoted as P˜h =Pee − Ph. Additionally, Pd ∈ R3 represents the position of the door. The positional difference between the end effector and the door is represented by P˜s=Pee − Pd. Moreover, Θi and Θh are the angles of the door’s hinge and handle, both expressed in radians. Second, in the block-stacking task, Sb=Pb,Qb ∈ R7  is comprised of the position and orientation of the target block. Here, Pb ∈ R3 represents the position of the target block, which is the objective for the first reaching task, and Qb∈ℝ4  denotes the orientation of the target block, which is expressed in quaternion form. P˜b=Pee − Pb means the positional difference between the end effector and the target block. Ps ∈ R3 denotes the position of another block that is designated for stacking, termed the base block. The difference in position between the end effector and base block is expressed as P˜s=Pee − Ps. The positional difference between the target block, which is the objective for the first reaching task, and the base block is denoted as P˜o=Pb − Ps. Finally, in the task of nut assembly, Sn=Pn,Qn ∈ R7 is a vector consisting of the nut’s current position Pn ∈ R3 and orientation Qn ∈ R4. The difference in position and orientation between the end effector and the nut is denoted by S˜n=See − Sn. Pnh ∈ R3 indicates the position of the nut handle, which is to be grasped next. The positional difference between the end effector and the nut handle is expressed as P˜nh=Pee − Pnh. Pp ∈ R3 is the position of the peg designed to fit the nut. The positional difference between the nut and the peg is represented by P˜p=Pn − Pp.

#### 3.4.2. Task-Specific Reward System for Reaching

The task-specific reward system for the reaching task is designed by considering the difference between the end effector’s position and the target position, as well as the energy consumption of the robot manipulator. It is defined as ${r=r}_{p}{+r}_{e}$, where
(9)$$r_{p}{=P}_{e}^{T\;}{\mathrm{KP}}_{e}{,\;P}_{e}{=P}_{\mathrm{ee}}{\;-\;P}_{t}=\left[\begin{array}{c} x_{\mathrm{ee}}{\;-\;x}_{t}\\ y_{\mathrm{ee}}{\;-\;y}_{t}\\ z_{\mathrm{ee}}{\;-\;z}_{t} \end{array}\right],\;K=\left[\begin{array}{ccc} K_{x} & 0 & 0\\ 0 & K_{y} & 0\\ 0 & 0 & K_{z} \end{array}\right]{,\;r}_{e}=-0.00003\;\times \;\overset{8}{\underset{i=1}{\sum }}{|F}_{i}|.$$

Here, Table 2 provides detailed descriptions of the target position and reward feedback parameters for each subtask. Considering the robot manipulator’s initial position, feedback parameters $K_{x}$, ${\;K}_{y}$ and $K_{z}$ are established to precisely reach the target destinations for each task. These parameters adjust the robot’s dynamic movements to achieve more precise targeting. All K feedback parameters are initially set intuitively to suit each task, and then through a process of trial and error, the parameters are adjusted to optimized values.

In the door-opening task, to reach the door handle, the most important factor is aligning the z-position of the end effector with that of the door handle, followed by aligning the y-position. Therefore, according to priority, the absolute values of the feedback parameters are set large in the order of $K_{z}$, $K_{y}$, and $K_{x}$, as shown in Table 2. This is essential for the gripper to properly grip the door handle. Additionally, in the block-stacking and nut assembly tasks, to increase the accuracy of arriving at the (x, y) position of the object, the values of the feedback parameters $K_{x}$ and $K_{y}$ are increased to be higher than $K_{z}$. Aligning the z position of the end effector is important, but more emphasis should be placed on matching the (x, y) position first in reaching the task. This is because precisely aligning the (x, y) position allows for accurate object grasping through the simple gripper action. Lastly, energy reward (9) is used as the actuator force $F_{i}$ that is measured at each joint through a simulation tool during the movement of the robot manipulator. 

#### 3.4.3. Task-Specific Reward System for Turning and Pulling

In this section, the task-specific reward system for the turning task is designed by considering the difference between the door handle angle and the target angle for unlocking. Similarly, for the pulling task, the task-specific reward system is defined utilizing the difference between the hinge angle and the target angle for pulling. As well as the energy consumption of the robot manipulator. It is defined as ${r=r}_{a}{+r}_{e}$, where $r_{a}{=K}_{o}{\left(\Theta_{o}\;-\;\Theta \right)}^{2}$.

Table 3 represents the target angle and feedback parameter according to the subtask. Depending on the task, the target angle is set to the unlock angle or the door-opening angle, and the feedback parameter $K_{o}$ is used the same for both tasks and set to the high value to increase sensitivity to angle changes. Also, the formula for the energy reward is the same as Equation (9).

#### 3.4.4. Task-Specific Reward System for Aligning

The task-specific reward system for the aligning task is designed by the orientation difference between the end effector and the nut, as well as the energy consumption of the robot manipulator. It is defined as ${r=r}_{o}{+r}_{e}$ where,
(10)$$r_{o}{=O}_{e}^{T\;}{\mathrm{KO}}_{e}{,\;O}_{e}{=O}_{\mathrm{ee}}{\;-\;O}_{n}=\left[\begin{array}{c} \theta_{\mathrm{ee}}{\;-\;\theta }_{n}\\ \phi_{\mathrm{ee}}\;-\;\phi_{n} \end{array}\right],\;K=\left[\begin{array}{cc} K_{\theta } & 0\\ 0 & {\;K}_{\phi } \end{array}\right]{,\;K}_{\theta }{=K}_{\phi \;}=-1.$$

The quaternion values in the state are converted to Euler form to set the reward. The orientation of the end effector and nut is aligned using the pitch and yaw values. In this subtask, the feedback parameters $K_{\theta }$ and $K_{\phi }$ are both equal to −1 (10). The formula for the energy reward is the same as Equation (9).

#### 3.4.5. Task-Specific Reward System for Assembly

The task-specific reward system for assembly task is designed by the positional difference between the nut and peg, as well as the energy consumption of the robot manipulator. It is defined as ${r=r}_{p}{+r}_{e}$ where,
(11)$$r_{p}{=P}_{e}^{T\;}{\mathrm{KP}}_{e}{,\;P}_{e}{=P}_{n}\;{-\;P}_{p}=\left[\begin{array}{c} x_{n}\;{-\;x}_{p}\\ y_{n}\;{-\;y}_{p}\\ z_{n}\;{-\;z}_{p} \end{array}\right],\;K=\left[\begin{array}{ccc} K_{x} & 0 & 0\\ 0 & K_{y} & 0\\ 0 & 0 & K_{z} \end{array}\right]{,\;K}_{x}{=K}_{y}{=-2,\;K}_{z}=-1.$$

In assembly tasks, it is important to prioritize the accuracy of the (x, y) positions not only for achieving precise peg-in-hole alignment between the nut and peg but also for facilitating the assembly process and enhancing efficiency. Therefore, to effectively perform assembly tasks, the feedback parameters $K_{x}$ and $K_{y}$ are set to absolute values higher than $K_{z}$ (11). The formula for the energy reward is the same as Equation (9).

## 4. Simulation Environment—Robosuite

Robosuite [28] is a framework that is designed for the development of algorithms in robot control and reinforcement learning. This framework provides a comprehensive set of tools for controlling robot systems. Additionally, it offers high compatibility with commonly used manipulators such as the Franka Emika Panda, Kinova3, Jaco, UR5, etc. One of the strengths of Robosuite is its provision of various gripper and object models. Users can utilize Robosuite to experiment and evaluate various control and reinforcement learning algorithms in different tasks such as block stacking, pick and place, and nut assembly. The flexible modular software architecture of Robosuite is designed to allow users to easily define and extend robots and their working environments, significantly enhancing the framework’s usability and applicability. Moreover, Robosuite supports a variety of machine learning techniques, not only reinforcement learning but also inverse reinforcement learning and imitation learning. These techniques facilitate the development and testing of robotic intelligence control algorithms, enabling users to address complex task environments and scenarios more effectively. 

## 5. Experimental Results

In this section, we present the experimental results aimed at deriving the optimal policies for each agent to successfully handle complex tasks such as door opening, block stacking, and nut assembly using the proposed method. The training process for all tasks was repeated four times. The training results are shown in figure, which averages the cumulative rewards for each episode obtained over four trials, and then shows the convergence of rewards and policy optimization trends through moving averages.

### 5.1. Door Opening

#### 5.1.1. Reaching Agent for Approaching the Door Handle

The first step in the door-opening task is to approach the door handle. The policy of the agent was established by the proposed reward system to achieve the goal of reaching the door handle from the initial position. The agent was trained for 3000 episodes with 300 steps per episode. Figure 5a illustrates that the reward converged to zero, indicating that the optimal policy for reaching the handle was established. The episode was considered a success if the end effector reached a certain range from the door handle, as follows:(12)$${\left(x_{\mathrm{ee}}\;{-\;x}_{h}\right)}^{2}\;<\;7{.5\;\times \;10}^{-5},\;{\left(y_{\mathrm{ee}}\;{-\;y}_{h}\right)}^{2}\;<\;7{.5\;\times \;10}^{-5},\;{\left(z_{\mathrm{ee}}\;{-\;z}_{h}\right)}^{2}\;<\;7{.5\;\times \;10}^{-5}.$$

The trained agent was tested for 500 episodes to reach the door handle, and these tests were repeated four times. The performance evaluation was found to be 99.95%, as shown in Table 4.

#### 5.1.2. Turning Agent for Unlocking

Rotating the handle was the second step in the door-opening task. The initial state of the agent was set to the information when the end effector grasped the door handle through the trained reaching agent and a simple gripper action. In this initial state, the agent was trained over 2000 episodes, with each episode consisting of 300 steps. Additionally, the agent was trained based on the proposed reward system, which aimed to minimize the difference between the current angle of the door handle and the angle required for unlocking. The convergence of the reward to the zero signified the establishment of the optimal unlocking policy, as shown in Figure 5b. Currently, success of the episode was defined as achieving the angle that released the lock. This criterion is represented as follows:(13)$${\left(\Theta_{h}\;-\;\frac{5}{12}\pi \right)}^{2}\;<\;0.0025.$$

The performance of trained agents was evaluated four times, with each evaluation consisting of 500 episodes. Through Table 5, it is demonstrated that each test achieved a 100% success rate, with an overall average success rate of 100%.

#### 5.1.3. Pulling Agent for Opening Door

Pulling the door open was the third step in the door-opening task. At this point, the initial state of training was set by using the first agent to reach the door handle and grasp it through a simple gripper action, and then using the second agent to turn the door handle, thereby unlocking it. In this initial state, the agent was trained across 2000 episodes, each consisting of 300 steps, based on a reward system for pulling the door. This reward system was designed to minimize the difference between the hinge angle and the angle required to open the door. Figure 5c represents the establishment of the optimal policy for pulling the door, as shown by the convergence of rewards to zero throughout the training process. Additionally, the episode was considered to have achieved its goal when the hinge angle reached the angle required for the door to open, with the following condition:(14)$$\;{\left(\Theta_{i}\;-\;\frac{1}{6}\pi \right)}^{2}\;<\;0.02.$$

To evaluate the pulling agent, the trained agent was tested four times, each consisting of 500 episodes. The average success rate was found to be 99.9%, as shown in Table 6. Also, since the door was opened through the pulling agent, the success rate of the pulling agent can be considered the success rate of the door-opening task. Consequently, the average success rate of the door-opening task was 99.9%.

### 5.2. Block Stacking

#### 5.2.1. Reaching Agent for Approaching the Target Block

The first objective of the block-stacking task was to reach the block, which is called a target block. The position of the target block was randomly generated within predefined boundaries, and its orientation was also randomly determined by z-axis rotation. The training of the agent, aimed at minimizing the positional difference between the end effector and the target block, spanned 5000 episodes, each comprising 300 steps. Figure 6a illustrates that the reward converged to zero, indicating that the optimal policy was established. Moreover, success was defined as the end effector accurately reaching the generated target block. This criterion is expressed as follows:
(15)$${\left(x_{\mathrm{ee}}\;{-\;x}_{b}\right)}^{2}\;<\;1{.25\;\times \;10}^{-4},\;{\left(y_{\mathrm{ee}}\;{-\;y}_{b}\right)}^{2}\;<\;1{.25\;\times \;10}^{-4},\;{\left(z_{\mathrm{ee}}\;{-\;z}_{b}\right)}^{2}\;<\;1{.75\;\times \;10}^{-4}.$$

The trained agent was evaluated four times, each consisting of 500 episodes. As indicated in Table 7, the average success rate for reaching the target block stood at 98.95%.

#### 5.2.2. Reaching Agent for Stacking the Block

Holding the target block and reaching over the base block in the target position was the second objective of the block-stacking task. In this case, the initial state was set when the end effector reached and grasped the target block by the first agent and simple gripper manipulation. From the initial state, the agent was trained across 5000 episodes, each consisting of 300 steps, utilizing the same reward system as the first reaching agent to minimize the positional difference between the target position and the end effector. Figure 6b shows the moving average trend initially fluctuating before gradually stabilizing, through which we can see that the reward converges to zero. In addition, the episode was considered a success when the end effector grasped the target block and reached the target position above the base block. This criterion is represented as follows:(16)$${\left(x_{\mathrm{ee}}\;{-\;x}_{s}\right)}^{2}\;<\;1{.25\;\times \;10}^{-4},\;{\left(y_{\mathrm{ee}}\;{-\;y}_{s}\right)}^{2}\;<\;1{.25\;\times \;10}^{-4},\;{\left(z_{\mathrm{ee}}\;{-\;z}_{s}\right)}^{2}\;<\;1{.75\;\times \;10}^{-4}.$$

The performance of the trained agent was evaluated in four trials with 500 episode tests to evaluate its performance in reaching the target position, resulting in an average success rate of 95.25%, as shown in Table 8.

Additionally, the block was stacked through the second agent, and the purpose of the putting task was to verify the state of the stacked blocks. Therefore, the success rate of the second agent and the putting task was the same, so the average success rate of the block-stacking task was 95.25%.

### 5.3. Nut Assembly

#### 5.3.1. Reaching Agent for Approaching Nut Handle Above

The first step of nut assembly was to approach the area above the nut handle. To achieve this, training for the agent spanned 2000 episodes, each consisting of 300 steps, and the agent was trained according to the proposed reward system. This system was designed to reach above the nut handle from the initial position by minimizing the difference in position between the target position and the end effector. Figure 7a illustrates the convergence of rewards to zero, which indicates the establishment of the optimal policy for reaching the target position. The performance of the trained agent was evaluated based on the success rate, which was determined by whether the end effector reached the target position according to the following criterion:(17)$${\left(x_{\mathrm{ee}}\;{-\;x}_{r}\right)}^{2}\;<\;1{.25\;\times \;10}^{-4},\;{\left(y_{\mathrm{ee}}\;{-\;y}_{r}\right)}^{2}\;<\;1{.25\;\times \;10}^{-4},\;{\left(z_{\mathrm{ee}}\;{-\;z}_{r}\right)}^{2}\;<\;1{.75\;\times \;10}^{-4}.$$

The trained agent’s ability to reach the target location was evaluated over 500 episodes four times. As shown in Table 9, the average success rates were 99.9% for the square nut and 99.875% for the round nut.

#### 5.3.2. Aligning Agent for Rotating Nut

To match the orientation of the end effector and the nut was the second step of nut assembly. The initial position of the end effector was set above the nut handle through the first agent. In this initial state, the agent was trained across 2000 episodes, each comprising 300 steps, with the goal of minimizing the difference in pitch and yaw between the end effector and the nut. Figure 7b illustrates that the reward converges to zero, which demonstrates that the optimal policy for orientation alignment was found. The episode was considered a success when the difference in pitch and yaw between the end effector and the nut satisfied the following conditions:(18)$${\left(\theta_{\mathrm{ee}}{\;-\;\theta }_{n}\right)}^{2}\;<\;0.01,\;{\left(\phi_{\mathrm{ee}}\;-\;\phi_{n}\right)}^{2}\;<\;0.001.$$

As represented in Table 10, the performance of the trained agent was evaluated over 500 episodes, repeated four times, demonstrating an average success rate of 97.45% for the square nut and 97.4% for the round nut, as indicated in Table 10.

#### 5.3.3. Reaching Agent for Approaching Nut Handle

The third step was to approach the nut handle. Initially, the end effector reached above the nut handle via the first agent, and then, its orientation with the nut was aligned by the second agent to establish the initial state. In this initial state, the agent was trained over 6000 episodes with 300 steps per episode according to the proposed reward system to reach the nut handle. In Figure 7c, it can be seen that the reward converges to zero, which means that the optimal policy for reaching the nut handle was established. The success of the episode was determined by whether the end effector reached the nut handle based on the following condition:(19)$${\left(x_{\mathrm{ee}}{\;-\;x}_{\mathrm{nh}}\right)}^{2}\;<\;1{.25\;\times \;10}^{-4},\;{\left(y_{\mathrm{ee}}{\;-\;y}_{\mathrm{nh}}\right)}^{2}\;<\;1{.25\;\times \;10}^{-4},\;{\left(z_{\mathrm{ee}}{\;-\;z}_{\mathrm{nh}}\right)}^{2}\;<\;1{.75\;\times \;10}^{-4}.$$

The performance of the trained agent to reach the nut handle was evaluated in 500 episodes, repeated four times. As a result, the average success rates were 95.7% for square nut and 98.45% for round nut, as indicated in Table 11.

#### 5.3.4. Assembly Agent for Peg in Hole

The fourth step of nut assembly was completing the peg-in-hole task with the nut. First, the end effector reached above the nut handle, aligned its orientation with the nut, and then reached the nut handle. This was accomplished using the first, second, and third agents, respectively. Subsequently, the nut handle was grasped through a simple gripper action. Using this state as the initial state, the agent was trained for 10,000 episodes, with 300 steps per episode, through the proposed reward system that was designed to perform peg-in-hole task with precision. Figure 7d shows that the reward converges to zero, indicating that the optimal policy was established. At this point, success was defined as precisely fitting the nut onto the peg and ensuring the difference was within the predefined threshold as follows:(20)$${\left(x_{n}{\;-\;x}_{p}\right)}^{2}\;<\;1{.25\;\times \;10}^{-4},\;{\left(y_{n}{\;-\;y}_{p}\right)}^{2}\;<\;1{.25\;\times \;10}^{-4},\;{\left(z_{n}{\;-\;z}_{p}\right)}^{2}\;<\;1{.75\;\times \;10}^{-4}.$$

The agent trained to accurately perform the peg-in-hole task was tested for 500 episodes, repeated four times. Finally, Table 12 shows the success rate and average success rate for each trial. Through this, the average success rate was 80.8% for square nut and 90.9% for round nut.

Furthermore, the peg-in-hole task was achieved through the fourth agent, and the purpose of the putting task was to verify the peg-in-hole state. So, since the success rate of the fourth agent and the putting task was the same, the average success rate for the square-nut assembly was 80.8%, and that for the round-nut assembly was 90.9%.

**Remark** **1.**
*As mentioned in the experimental results, except for the first task, subsequent tasks used the agent that was trained in the previous subtask to create the initial state. For example, in the door-opening task, the process of reaching the door handle was implemented using a reaching agent, and the door handle was grasped through a simple gripper action. This state of holding the door handle was then used as the initial state for the turning agent. Another example is in the nut assembly task, where the first reaching agent was used to reach above the nut handle, and an aligning agent was used to align the orientation of the end effector with the nut. Subsequently, the nut handle was reached using the second reaching agent. The nut handle was grasped through a simple gripper action. The state of holding the nut handle was used as the initial state for the assembly agent. Thus, for tasks following the first, initial states were set using agents from the previous subtask.*


## 6. Discussion

Through the proposed method of this research, high success rates of 99.9%, 95.25%, 90.9%, and 80.8% were achieved in door-opening, block-stacking, and round- and square-nut assembly tasks, respectively. As a result, it can be considered that higher performance was achieved compared to the benchmarking results of Robosuite. In particular, the Robosuite benchmarking results for the block stacking and nut assembly tasks showed poor learning performance. However, applying the proposed method for the same tasks resulted in achieving high accuracy and learning rates. Moreover, comparing the proposed method to the excellent performance of the door-opening task in Robosuite indicates that the proposed method obtained similar or even superior results. This suggests that the proposed method exhibits strengths not only in complex tasks but also in less complex tasks. This demonstrates that the proposed method is versatile and effective in performing complex and diverse tasks.

However, there are some limitations to the methodology of this study. Unlike round-nut assembly, achieving precise peg-in-hole fitting in square-nut assembly tasks requires setting a reward that considers the orientation of the square nut. However, by setting it to the same as that for the round nut, the success rate was relatively lower compared to the round-nut assembly task. In other words, the proposed method has the advantage of improving the performance of the reinforcement learning algorithm by designing a reward system for a specific task. However, it may be sensitive to changes in the environment of the task, making it difficult to generalize the reward system. Additionally, the approach of sequentially connecting optimal policies to perform complex tasks has the problem that failures of subtasks accumulate and affect the final performance.

To overcome these limitations, future research could consider training additional agents that effectively connect optimal policies for subtasks. The objective is to increase the overall task success rate and enhance flexibility and adaptability in performing complex tasks. This can be achieved by establishing advanced policies that efficiently integrate and connect tasks, rather than simply sequentially connecting policies for each subtask. Through future research, it is possible to prevent the negative impact on overall performance caused by failures in subtasks while simultaneously enhancing flexibility and adaptability, thereby further improving the ability to perform complex tasks. 

## 7. Conclusions

This paper proposed a reinforcement learning method based on task decomposition and a task-specific reward system for performing complex high-level tasks such as door opening, block stacking, and nut assembly. Initially, the door-opening task was decomposed into subtasks of reaching–grasping–turning–pulling. The block-stacking task was divided into subtasks of reaching–grasping– reaching–putting, and the nut assembly task was divided into subtasks of reaching–aligning–reaching–grasping–assembling–putting. The grasping and putting tasks were implemented with single joint and gripper actions, while agents for the other tasks were trained using the SAC algorithm and a task-specific reward system. Here, the task-specific reward system was used to increase the learning speed of the agents, enhance their success rates, and facilitate the subsequent tasks of grasping and putting more efficiently. The proposed method was validated by successfully completing tasks with a 99.9% success rate for door opening, 95.25% for block stacking, 90.9% for round-nut assembly, and 80.8% for square-nut assembly. This demonstrates a significant improvement over existing methods in performing complex tasks. By overcoming the limitations of end-to-end approaches, it proves to be a valuable solution for solving diverse and complex tasks while also presenting new directions for future research.

## Figures and Tables

**Figure 1 biomimetics-09-00196-f001:**
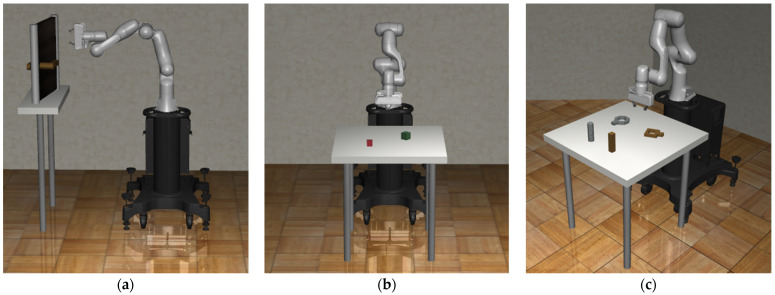
The initial state of each task provided by Robosuite: (**a**) door-opening task; (**b**) block-stacking task; (**c**) nut assembly task.

**Figure 2 biomimetics-09-00196-f002:**
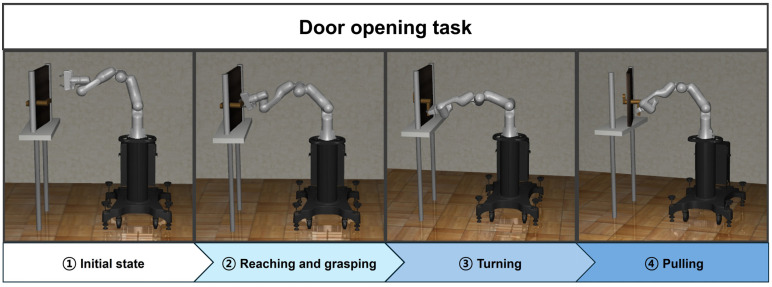
The overall workflow of the proposed method for the door-opening task.

**Figure 3 biomimetics-09-00196-f003:**
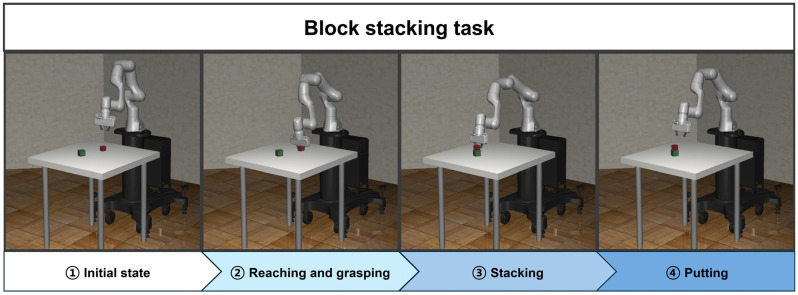
The overall workflow of the proposed method for the block-stacking task.

**Figure 4 biomimetics-09-00196-f004:**
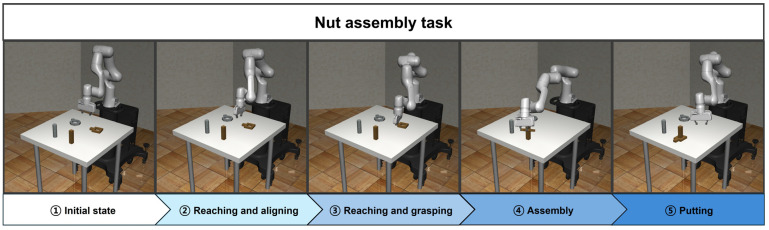
The overall workflow of the proposed method for the nut assembly task.

**Figure 5 biomimetics-09-00196-f005:**
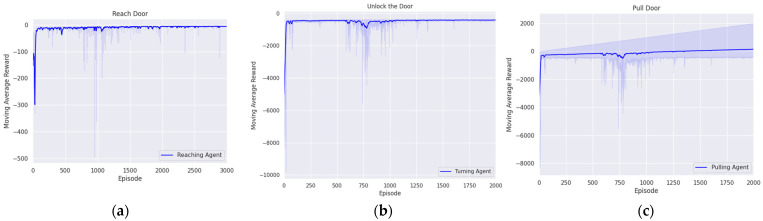
Trend in moving average reward during door-opening task across episodes: (**a**) reaching agent; (**b**) turning agent; (**c**) pulling agent.

**Figure 6 biomimetics-09-00196-f006:**
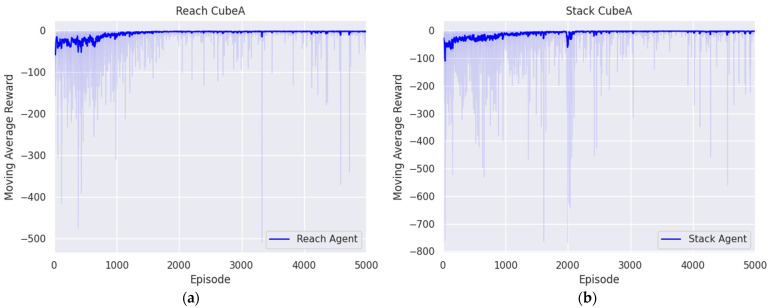
Trend in moving average reward during block-stacking task across episodes: (**a**) first reaching agent; (**b**) second reaching agent.

**Figure 7 biomimetics-09-00196-f007:**
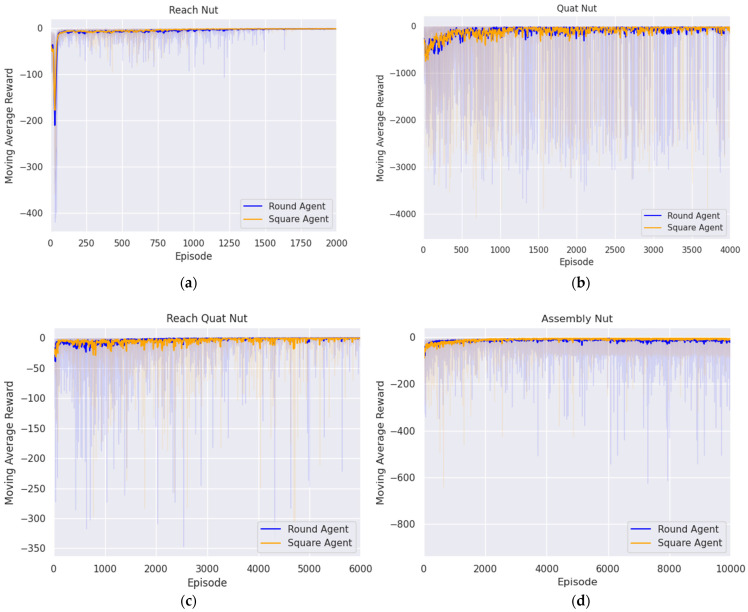
Trend in moving average reward during nut assembly task across episodes: (**a**) first reaching agent; (**b**) aligning agent; (**c**) second reaching agent; (**d**) assembly agent.

**Table 1 biomimetics-09-00196-t001:** This table represents the state of the robot manipulator commonly used for all tasks and the state of the object used for each task.

Posture of the Robot Manipulator		
$S_{\mathrm{ee}}$: position and orientation of the end effector $\Theta_{\cos}$: cosine values of the joint angles $\Theta_{\sin}$: sine values of the joint angles $\Theta_{\mathrm{vel}}$: joint velocity $S_{g}$: position and velocity of the gripper		
Door opening	Block stacking	Nut assembly
$P_{h}$: door handle position ${\tilde{P}}_{h}$: positional difference between handle and end effector $P_{d}$: door position ${\tilde{P}}_{d}$: positional difference between door and end effector $\Theta_{i}$: hinge angle $\Theta_{h}$: handle angle	$S_{b}$: position and orientation of the target block ${\tilde{P}}_{b}$: positional difference between target block and end effector $P_{s}$: position and orientation of the base block ${\tilde{P}}_{s}$: positional difference between base block and end effector ${\tilde{P}}_{o}$: positional difference between target block and base block	$S_{n}$: position and orientation of the nut ${\tilde{S}}_{n}$: positional difference between nut and end effector $P_{\mathrm{nh}}$: nut handle position ${\tilde{P}}_{\mathrm{nh}}$: positional difference between nut handle and end effector $P_{p}$: peg position ${\tilde{P}}_{p}$: positional difference between nut and peg

**Table 2 biomimetics-09-00196-t002:** This table indicates that the target position and reward feedback parameters vary for each subtask within the reaching task.

Subtask	Target Position	$K_{x}$	$K_{y}$	$K_{z}$
Reaching task in door opening	$P_{h}$: door handle	−0.5	−2	−4
First reaching task in block stacking	$P_{b}$: block	−2	−2	−1
Second reaching task in block stacking	$P_{s}$: above target block	−2	−2	−1
First reaching task in nut assembly	$P_{r}$: above nut handle	−2	−2	−1
Second reaching task in nut assembly	$P_{\mathrm{nh}}$: nut handle	−2	−2	−1

**Table 3 biomimetics-09-00196-t003:** The target angle and reward feedback parameters for each subtask within the turning and pulling task.

Subtask	$\Theta_{o}$	Target Angle Θ	$K_{o}$
Turning task in door opening	$\Theta_{h}$: door handle angle	$\frac{5}{12}\pi$	−10
Pulling task in door opening	$\Theta_{i}$: hinge angle	$\frac{1}{6}\pi$	−10

**Table 4 biomimetics-09-00196-t004:** This is the success rate for each trial and the average success rate for all trials. The test process involved the robot manipulator reaching the door handle using the established policy.

Trial 1	Trial 2	Trial 3	Trial 4	Average
100%	100%	100%	99.8%	99.95%

**Table 5 biomimetics-09-00196-t005:** This is the success rate for each trial and the average success rate for all trials. The test process involved the robot manipulator turning the door handle using the established policy.

Trial 1	Trial 2	Trial 3	Trial 4	Average
100%	100%	100%	100%	100%

**Table 6 biomimetics-09-00196-t006:** This is the success rate for each trial and the average success rate for all trials. The test process involved the robot manipulator pulling the door using the established policy.

Trial 1	Trial 2	Trial 3	Trial 4	Average
99.6%	100%	100%	100%	99.9%

**Table 7 biomimetics-09-00196-t007:** This is the success rate for each trial and the average success rate for all trials. The test process involves the robot manipulator reaching the block using the established policy.

Trial 1	Trial 2	Trial 3	Trial 4	Average
99.0%	99.2%	98.8%	98.8%	98.95%

**Table 8 biomimetics-09-00196-t008:** This is the success rate for each trial and the average success rate for all trials. The test process involved the robot manipulator stacking the block using the established policy.

Trial 1	Trial 2	Trial 3	Trial 4	Average
96.0%	95.2%	94.6%	95.2%	95.25%

**Table 9 biomimetics-09-00196-t009:** This is the success rate for each trial and the average success rate for all trials. The test process involved the robot manipulator reaching above the nut handle using the established policy.

	Trial 1	Trial 2	Trial 3	Trial 4	Average
Square	99.8%	100%	100%	99.8%	99.9%
Round	99.9%	99.6%	100%	100%	99.875%

**Table 10 biomimetics-09-00196-t010:** This is the success rate for each trial and the average success rate for all trials. The test process involved the robot manipulator aligning the nut using the established policy.

	Trial 1	Trial 2	Trial 3	Trial 4	Average
Square	98.6%	97.6%	97.0%	98.0%	97.45%
Round	97.2%	97.8%	96.6%	98.0%	97.4%

**Table 11 biomimetics-09-00196-t011:** This is the success rate for each trial and the average success rate for all trials. The test process involved the robot manipulator reaching the nut handle using the established policy.

	Trial 1	Trial 2	Trial 3	Trial 4	Average
Square	96.0%	95.0%	94.2%	97.6%	95.7%
Round	98.8%	97.6%	99.0%	98.4%	98.45%

**Table 12 biomimetics-09-00196-t012:** This is the success rate for each trial and the average success rate for all trials. The test process involved the robot manipulator assembly of the nut using the established policy.

	Trial 1	Trial 2	Trial 3	Trial 4	Average
Square	80.4%	81.8%	80.2%	80.8%	80.8%
Round	89.8%	90.8%	91.0%	92.0%	90.9%

## Data Availability

Data are contained within the article.

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
