# Peer review of "Reinforcement Learning with Task Decomposition and Task-Specific Reward System for Automation of High-Level Tasks"

_biomimetics, 2024, doi:10.3390/biomimetics9040196_

Round 1

Reviewer 1 Report

Comments and Suggestions for Authors

The paper titled with “Reinforcement Learning with Task Decomposition and Task-Specific Reward System for Automation of High-level Tasks” presents application of reinforcement learning method for door opening, block stacking, and nut assembly. The proposed method was validated by successfully completing tasks with a 99.9% success rate for door opening, 95.25% for block stacking, 90.9% for round nut assembly, and 80.8% for square nut assembly. This demonstrates that the proposed method overcomes the limitations of the end-to-end approach and represents a useful solution for solving various complex and challenging tasks. Although the study covers a general robotic manipulation problem, it reports increased success rate with reinforcement learning. The manuscript is well written, however following minor comments must be addressed before publication.

1.                Improve clarity for the figure 5.

2.       Some recent related literatures on the reinforcement learning in automation.

3.         The descriptions of future perspectives should end the section before conclusion. And following paragraph should be deleted or move to the conclusion.

In conclusion, this research demonstrates that the proposed method has achieved significant improvements over existing methods in performing complex tasks, while also suggesting new directions for future research.”

Reviewer 2 Report

Comments and Suggestions for Authors

The paper presents a reinforcement learning approach for task decomposition and especially a task-specific reward system to solve several challenges regarding robotic manipulation.

The paper is well-written and very readable.

The task decomposition approach is not especially innovative, but the paper provides extensive description about how to apply it in robotic manipulation environments.

In line 257, authors mention “state” and “action”, but maybe they refer to “state space” and “action space”? I think the same applies throughout all Section 3.4.1, please revise.

The task specific reward system, which is one of the main contributions of the paper, is well presented and adequate to easily generalize and prioritize some elements of the reward formulation based on their importance for the performance of the task. No much information is presented about how the authors obtained the values of the different K feedback parameters (intuition, trial and error, …), maybe some explanation could be interesting here.

All the experiments and trials of Section 4 are well-defined and documented, especially important are the success criteria for considering the task is completed, which the authors provide.
